# Towards Unbiased Calibration using Meta-Regularization

**Cheng Wang**[†]                                                                      *cwngam@amazon.com*
*Amazon*
*Berlin, Germany*

**Jacek Golebiowski**[†]                                                               *jacekgo@amazon.com*
*Amazon*
*Berlin, Germany*

**Reviewed on OpenReview:** *https://openreview.net/forum?id=Yf8iHCfG4W*

## Abstract

Model miscalibration has been frequently identified in modern deep neural networks. Recent work aims to improve model calibration directly through a differentiable calibration proxy. However, the calibration produced is often biased due to the binning mechanism. In this work, we propose to learn better-calibrated models via meta-regularization, which has two components: (1) gamma network ($\gamma$-Net), a meta learner that outputs sample-wise gamma values (continuous variable) for Focal loss for regularizing the backbone network; (2) smooth expected calibration error (SECE), a Gaussian-kernel based, unbiased, and differentiable surrogate to ECE that enables the smooth optimization of $\gamma$-Net. We evaluate the effectiveness of the proposed approach in regularizing neural networks towards improved and unbiased calibration on three computer vision datasets. We empirically demonstrate that: (a) learning sample-wise $\gamma$ as continuous variables can effectively improve calibration; (b) SECE smoothly optimizes $\gamma$-Net towards unbiased and robust calibration with respect to the binning schemes; and (c) the combination of $\gamma$-Net and SECE achieves the best calibration performance across various calibration metrics while retaining very competitive predictive performance as compared to multiple recently proposed methods.

## 1 Introduction

Deep Neural Networks (DNNs) have demonstrated promising predictive performance in various domains, including computer vision (Krizhevsky et al., 2012), speech recognition (Graves et al., 2013), and natural language processing (Vaswani et al., 2017). Consequently, trained deep neural network models are frequently deployed and utilized in real-world systems. However, recent work (Guo et al., 2017) has pointed out that these highly accurate, negative log likelihood-trained deep neural networks are often poorly calibrated (Niculescu-Mizil & Caruana, 2005b). Their predicted class probabilities do not accurately estimate the true probability of correctness, resulting in primarily overconfident and under-confident predictions. Deploying such miscalibrated models in real-world systems poses significant risk, particularly when model outputs are directly utilized to serve customers' requests in applications such as medical diagnosis (Caruana et al., 2015) and autonomous driving (Bojarski et al., 2016). Better-calibrated model probabilities can serve as an essential signal toward building more reliable machine learning systems.

A recent trend aims to learn a calibrated model by training it to minimize errors on calibration metrics. One representative work is from Kumar et al. (2018), where they developed a differentiable equivalent of Expected Calibration Error (ECE), the Maximum Mean Calibration Error (MMCE). Mukhoti et al. (2020) found that focal loss (Lin et al., 2017), as a good alternative to standard cross-entropy, can effectively improve calibration. They also highlighted the crucial role of the gamma parameter in focal loss in making this approach

---

[†]Equal contribution.

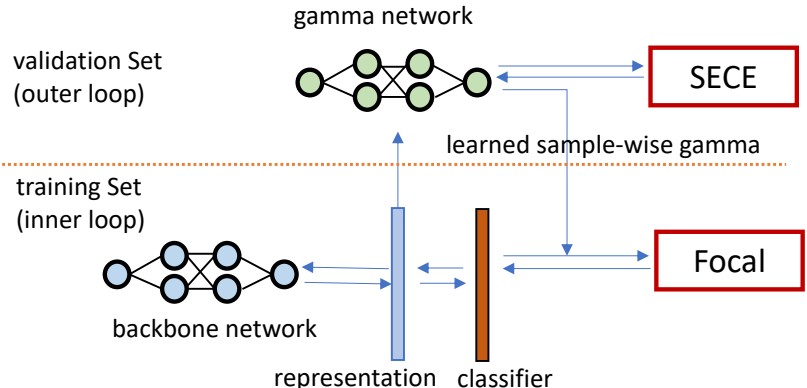

Figure 1: Our proposed approach for regularizing the base network towards better calibration includes two new components: $\gamma$-Net and SECE . The inner loop optimizes the backbone network (e.g., ResNet), which uses focal loss as an objective function. The $\gamma$-Net in the outer loop takes the extracted second-to-last layer representation of backbone network as input and learns to output sample-wise $\gamma$ for focal loss in a continuous space. The $\gamma$-Net is optimized by using the proposed SECE, a Gaussian kernel-based, unbiased, and differentiable calibration error.

effective and proposed a sample-dependent schedule based on heuristics (Lambert-W function (Corless et al., 1996)) for the gamma in focal loss (FLSD), which showed superior calibration performance compared to baselines. Bohdal et al. (2021; 2023) proposed using a meta-learning-based approach termed as meta-calibration. In their formulation, the backbone network learns to optimize a standard cross-entropy loss, while a differentiable proxy for calibration error (i.e., DECE) is used to tune the parameters of the weight regularizer.

Motivated by advancements in this area, we introduce a learnable approach that involves learning sample-wise $\gamma$ values via a meta-network, referred to as the gamma network (i.e., $\gamma$-Net), as illustrated in Figure 1. The key distinctions from Focal loss (Lin et al., 2017) and FLSD (Mukhoti et al., 2020), where a global $\gamma$ parameter (e.g., setting $\gamma = 3$ or $\gamma = 5$) is employed as a fixed parameter or scheduled across training epochs, are twofold: our approach (1) learns more finely-grained gamma values at the sample level; (2) the learned $\gamma$ values are continuous variables. The output of the $\gamma$-Net, namely the learned $\gamma$ values, is then utilized to regularize a backbone network towards improved calibration. The optimization of the $\gamma$-Net is achieved with a proposed differentiable surrogate for expected calibration error (ECE) (Naeini et al., 2015): Smooth ECE (SECE), an unbiased Gaussian kernel based ECE estimator. SECE optimizes $\gamma$-Net towards unbiased calibration by avoiding the binning step as in (Zhang et al., 2020).

Our contributions can be summarized as follows:

- We propose $\gamma$-Net, a meta-network designed to learn sample-wise $\gamma$ values for Focal loss as continuous variables, rather than relying on a single pre-defined $\gamma$ value. To the best of our knowledge, our proposed method learns the most fine-grained $\gamma$ values for Focal loss.

- We propose a kernel-based ECE estimator,SECE , which is a variation of the kernel density estimate (KDE) approximation (Zhang et al., 2020). This estimator smoothly regularizes $\gamma$-Net towards improved and unbiased calibration.

- Through extensive experiments and analysis, we empirically demonstrate that $\gamma$-Net effectively calibrates models, while SECE provides stable and smooth calibration. The combination of both achieves competitive predictive performance and superior scores across multiple calibration metrics compared to the baselines.

## 2   Related Work

Calibration of machine learning (ML) models using Platt scaling (Platt et al., 1999) and Isotonic Regression (Zadrozny & Elkan, 2002) has shown significant improvements for SVMs and decision trees. With the advent of neural networks, Niculescu-Mizil & Caruana (2005a) showed that those methods can produce well-calibrated probabilities even without any dedicated modifications.

Recently, Guo et al. (2017) and Mukhoti et al. (2020) have shown that modern NNs are noticeably more accurate but poorly calibrated due to negative log likelihood (NLL) overfitting. Minderer et al. (2021) revisited this problem and found that architectures with large sizes have a larger effect on calibration, but nevertheless, more accurate models tend to produce less calibrated predictions.

One of the ways to tackle calibration is to rely on post-hoc corrections using non-parametric approaches such as histogram binning (Zadrozny & Elkan, 2001), isotonic regression (Zadrozny & Elkan, 2002) and parametric methods such as Bayesian binning into quantiles (BBQ) and Platt scaling (Platt et al., 1999). Beyond those four, temperature scaling (TS) is a single-parameter extension of Platt scaling (Platt et al., 1999) and the most recent addition to the offering of post-hoc methods. Recent work has shown that a single parameter TS leads to good calibration performance with minimal added computational complexity (Guo et al., 2017; Minderer et al., 2021). There have been multiple extensions to temperature scaling in recent years. Kull et al. (2019) proposed Dirichlet calibration which assumes probability distributions are parameterized Dirichlet distributions, and an attention-based mechanism was proposed to tackle noise in validation data (Mozafari et al., 2018). Laves et al. (2019) extended TS to dropout variational inference to calibrate model uncertainty by inserting temperature value in Monte Carlo integration.

Beyond post-hoc methods, other approaches do not explicitly include the calibration objective but rather implicitly guide the training towards better calibration performance. There are basically two lines of work: (1) data augmentation-based methods and (2) regularization-based methods. For the former line, the representative methods include label smoothing (LS) (Müller et al., 2019) that mitigates miscalibration by softening hard labels with an introduced smoothing modifier in the standard loss function (e.g., cross-entropy). Mix-up training (Thulasidasan et al., 2019) that extends *mixup* (Zhang et al., 2018) to generate synthetic samples during model training by combining two random elements from the dataset.

Extending the portfolio of regularization-based methods for calibration, focal loss (Lin et al., 2017) has been used as a maximum entropy regularizer. One of its extensions, FLSD (Mukhoti et al., 2020) computes the gamma value for focal loss based on heuristics (Lambert-W function (Corless et al., 1996)), dual Focal (Tao et al., 2023) takes into account the logit corresponding to the ground truth label and the largest logit ranked after it. AdaFocal (Ghosh et al., 2022) utilizes the calibration properties of focal (and inverse-focal) loss and adaptively modifies gamma for different groups of samples based on previous gamma value the knowledge of model's under/over-confidence on the validation set. Most recently, Krishnan & Tickoo (2020) introduced a differentiable accuracy versus uncertainty calibration (AvUC) loss function that allows a model to learn to provide well-calibrated uncertainties. Liu et al. (2022) analyzed some current calibration methods such as label smoothing (Müller et al., 2019), focal loss (Lin et al., 2017), and explicit confidence penalty (Pereyra et al., 2017) from a constrained-optimization perspective and pointed out those can be seen as approximations of a liner penalty. The authors proposed to add a margin into learning objectives. Cheng & Vasconcelos (2022) proposed calibration by pairwise constraints (CPC), which aims to strengthen the calibration supervision by casting multiclass searino into pairwise binary calibration. Thus authors proposed binary discrimination constraints (BDC) loss and binary exclusion constraint (BEC) loss as addition to standard cross-entropy loss to increase the supervision rate for calibration of the training process.

Within regularization-based methods, some approaches use differentiable calibration proxies as regularizers. For instance, Kumar et al. (2018) developed a differentiable equivalent of ECE, the Maximum Mean Calibration Error (MMCE), which can be optimized directly to train calibrated models. Similarly Karandikar et al. (2021) proposed a softened version of ECE (SB-ECE) by employing a softmax-based bin-membership function with temperature, enabling the soft ECE to serve as either a primary or auxiliary loss objective. Another interesting exploration is presented in Bohdal et al. (2021), where instead of utilizing a developed differentiable ECE (DECE) for optimizing the main network, the authors proposed a meta-learning

approach (Luketina et al., 2015) to train a model that minimizes DECE, thereby finding the optimal parameters of the L2 regularizers.

Compared to the aforementioned regularization methods, to the best of our knowledge, our proposed method learns the most fine-grained gamma values for focal loss via a meta-network. This network is optimized toward stable and unbiased calibration, mitigating the biases caused by the binning mechanism.

## 3 Preliminaries

### 3.1 Model Calibration

Calibration (Guo et al., 2017) measures and verifies how well the predicted probability estimates the true likelihood of correctness. Assume a model $\mathcal{M}$ trained with dataset $\{\mathbf{x}, y\}, \mathbf{x} \in \mathcal{X}, y \in \mathcal{Y}$. Let $\mathbf{p} = \{p_1, p_2, ..., p_K\}$ be the predicted softmax probability of $K$ classes. If $\mathcal{M}$ makes 100 independent predictions, each with confidence $p = \arg\max(\mathbf{p}) = 0.9$, a calibrated $\mathcal{M}$ is correct 90 times. Formally, $\text{accuracy}(\mathcal{M}(D)) = \text{confidence}(\mathcal{M}(D))$ if $\mathcal{M}$ is perfectly calibrated on dataset $D$.

**Reliability Diagrams** (DeGroot & Fienberg, 1983; Niculescu-Mizil & Caruana, 2005a) visualize whether a model is overconfident or underconfident by grouping predictions into bins according to their prediction probability. The predictions are grouped into $M$ bins $\{b_1, b_2, .., b_M\}$. The accuracy of samples in each bin $b_m$ is computed as: $\text{ACC}(b_m) = \frac{1}{|b_m|} \sum_{i \in b_m} \mathbf{1}(\hat{y}_i = y_i)$, where $i$ indexes all examples that fall into bin $b_m$, $\hat{y}_i$ are $y_i$ are the prediction and ground-truth, respectively. Let $\hat{p}_i$ be the probability for the predicted class $y_i$ for the $i$-th sample (i.e. the class with highest predicted probability for $i$-th sample), then the average confidence is defined as: $\text{CONF}(b_m) = \frac{1}{|b_m|} \sum_{i \in b_m} \hat{p}_i$. A model is perfectly calibrated if $\text{ACC}(b_m) = \text{CONF}(b_m), \forall m$ and in a diagram the bins would follow the identity function. Any deviation from this indicates miscalibration.

**Expected Calibration Error (ECE)** (Naeini et al., 2015). ECE computes the difference between model accuracy and confidence. It quantifies the discrepancy between predicted probabilities and observed frequencies by partitioning data points into bins of similar confidence. It takes the form

$$\text{ECE} = \sum_{m=1}^{M} \frac{|b_m|}{N} |\text{ACC}(b_m) - \text{CONF}(b_m)|, \tag{1}$$

where $N$ is the total number of samples and $b_m$ represents a single bin.

**Maximum Calibration Error (MCE)** (Naeini et al., 2015) is particularly important in high-risk applications where reliable confidence measures are absolutely necessary. It measures the worst-case deviation between accuracy and confidence, $\text{MCE} = max_{m \in \{1, ..., M\}} |\text{ACC}(b_m) - \text{CONF}(b_m)|$. For a perfectly calibrated model, the ideal ECE and MCE are equal to 0.

Besides ECE and MCE, we also report *Classwise ECE* (Nixon et al., 2019), *Adaptive ECE* (Nixon et al., 2019), we describe these metrics in Appendix A.1.

### 3.2 Focal Loss

For a classification task, the focal loss (FL) (Lin et al., 2017) can be defined as $\mathcal{L}_\gamma^f = -(1 - p_{i,y_i})^\gamma \log p_{i,y_i}$ where $\gamma$ is a hyper-parameter. It was originally proposed to handle imbalanced data distributions but Mukhoti et al. (2020) found that the models trained with focal loss are better calibrated with respect to cross-entropy trained counterparts. This is because focal loss can be interpreted as a trade-off between minimizing Kullback–Leibler (KL) divergence and maximizing the entropy of predictions, depending on $\gamma$ (Mukhoti et al., 2020)[†]:

$$\mathcal{L}_f \geq \text{KL}(q \parallel p) + \mathbb{H}(q) - \gamma \mathbb{H}(p) \tag{2}$$

where $p$ is the prediction distribution, $q$ is the one-hot encoded target class, and $\mathbb{H}(q)$ is constant. Models trained with this loss learn to strike a balance between narrowing $p$ (high confidence in predictions) due to

---

[†]More theoretical findings can be found in the paper.

the KL term and boarding $p$ (avoiding overconfidence) due to the entropy regularization term. The authors provided a principled approach to select the $\gamma$ for focal loss based on the Lambert-W function (Corless et al., 1996). Motivated by this, our work proposes to learn a more fine-grained, sample-wise $\gamma$ with a meta-network.

### 3.3 Meta-Learning

Model calibration can be formulated as minimizing a multi-component loss objective where the base term is used to optimize predictive performance while a regularizer maintains model calibration (Bohdal et al., 2021; Lin et al., 2017). Tuning the hyper-parameters of the regularizer can be a difficult process when conventional methods are used (Li et al., 2016; Snoek et al., 2012; Falkner et al., 2018). In this work, we adopt the meta learning approach (Luketina et al., 2015). At each training iteration, the model takes a mini-batch from the training dataset $D_{train}$ and the validation dataset $D_{val}$ to optimize

$$\arg\min_{\theta,\phi} \ \mathcal{L}(\theta, \phi, D_{train}, D_{val}) = \mathcal{L}_{FL_\gamma}(\theta, D_{train}) + \mathcal{L}_{SECE}(\phi, D_{val}) \tag{3}$$

First, the base loss function $\mathcal{L}_{FL_\gamma}$ is used to optimize the parameter of the backbone model $\theta$ on $D_{train}$, note that the $\gamma$ parameter for this step is predicted by $\gamma$-Net. Following that, the validation mini-batch $D_{val}$ is used to optimize the parameters of the meta-network ($\gamma$-Net) $\phi$ using a validation loss $\mathcal{L}_{SECE}$. The validation loss is a function of the backbone model output and does not depend on $\gamma$-Net directly. The dependence between the validation loss and parameters of the meta-network is mediated via the parameters of the backbone model as discussed by (Luketina et al., 2015).

## 4 Methods

This section introduces the two components of our approach: the $\gamma$-Net learns to parameterise the focal loss and the SECE provides differentiability to $\gamma$-Net towards calibration optimization.

### 4.1 $\gamma$-Net: Learning Sample-Wise Gamma for Focal Loss

Sample-dependent $\gamma$ value for focal loss showed it effectiveness of calibrating deep neural networks (Mukhoti et al., 2020). In this work, instead of computing and scheduling $\gamma$ value based on Lambert-W function (Corless et al., 1996). We propose a learnable approach which learns more fine-grained and local $\gamma$ values, i.e., each sample has an individual $\gamma$, in a continuous space. Formally, $\gamma$-Net takes the representations from the second-to-last layer of a backbone network (i.e. ResNet (He et al., 2016). Let $\mathbf{x} \in \mathbb{R}^{b \times d}$ ($b$: batch size, $d$: hidden dimension) be the extracted representation, $\mathbf{A} \in \mathbb{R}^{d \times k}$ be a $k$-head self-attention matrix that followed by a liner layer with parameters $\mathbf{W} \in \mathbb{R}^{d \times 1}$. The $\gamma$-Net transforms the representation to sample-wise $\gamma$:

$$\mathbf{a} = \mathbf{x} \cdot \mathbf{A}, \ \in \mathbb{R}^{b \times k}, \ \mathbf{p} = \text{SOFTMAX}(\mathbf{a}), \ \in \mathbb{R}^{b \times k} \tag{4}$$

$$\tilde{\mathbf{x}} = \mathbf{p} \cdot \mathbf{A}^\top, \ \in \mathbb{R}^{b \times d}, \ \gamma = |\tilde{\mathbf{x}} \cdot \mathbf{W}| / \tau, \ \in \mathbb{R}^{b \times 1} \tag{5}$$

Here we use $|\cdot|$ to ensure the $\gamma$ is positive valued and tune the temperature $\tau = 0.01$ as a hyperparameter. This is similar to the temperature setups as in meta-calibration (Bohdal et al., 2021; 2023). Those operations result in a set of sample-wise $\gamma$ to be used in focal loss $\mathcal{L}_r^f$. Note that, $f_\gamma(\gamma_i | x_i), x_i \in D$ is learned as a continous variable rather than discrete value as in original FL formulation (Lin et al., 2017) and scheduled FL (FLSD) (Mukhoti et al., 2020).

**Practical Considerations**: To ensure $\gamma \geq 0$, we could apply operations based on min-max scaling or activation functions such as sigmoid, ReLu(Rectified Linear Unit) Agarap (2018) and softplus. However, our experimental evidence shows that the formulation presented in Equation 5 performs better across datasets.

### 4.2 SECE : Smooth Expected Calibration Error

Conventional calibration measurement via Expected Calibration Error (ECE) discussed in Section 3.1, is computed over discrete bins by finding the accuracy and confidence of examples in each interval. However,

this approach is highly dependent on different settings of bin edges and the number of bins, making it a biased estimator of the true value (Minderer et al., 2021).

The issue of bin-based ECE can be traced back to the discrete decision of binning of samples (Minderer et al., 2021). The larger bin numbers, the less information is retained. Small bins lead to inaccurate measurements of accuracy. For instance, in single-example bins, the confidence is well defined but the accuracy in that interval of model confidences is more difficult to assess accurately from a single point. Using the binary accuracy of a single example leads us to the Brier score (Brier et al., 1950) which has been shown to not be a perfect measure of calibration.

To find a good representation of accuracy within the single-example bin (representing a small confidence interval), we leverage the accuracy of other points in the vicinity of the single chosen example weighted by their distance in the confidence space. Formally, the soft estimation of accuracy within a single-example bin:

$$\text{SACC}\ (b_i) = \sum_{j}^{M} \pi(x_i) K(z_i, z_j) \tag{6}$$

$$K\left(x_i, x'_j\right) = \exp\left(-\frac{\left\|x_i - x'_j\right\|^2}{2h^2}\right) \tag{7}$$

where $b_i$ is the bin housing example $x_i$ and $K(\cdot, \cdot)$ is a chosen distance measure, for example a Gaussian kernel and $h$ is the bandwidth. Moreover, $z_i$ and $\pi(x_i)$ respectively denote the confidence and accuracy of the $i-th$ example. As in meta-calibration (Bohdal et al., 2021), we use the all-pairs approach (Qin et al., 2010) to ensure differentiability. Having good measures of soft accuracy and confidence for each single-example bin, we can write the updated ECE metric as soft-ECE in the form

$$\text{SECE}\ = \frac{1}{M} \sum_{i}^{M} |\text{SACC}\ (i) - \text{CONF}(i)|, \tag{8}$$

where $i$ represents a single example and $M$ is the number of examples. The new formulation is (1) differentiable as long as the kernel we use is differentiable and (2) enables smooth control over how accuracy over a confidence bin is computed via tuning the kernel parameters. Regarding (2), choosing a Dirac-delta function recovers the original Brier score while choosing a broader kernel enables a smooth approximation of accuracy over all confidence values.

**Connection to KDE-based ECE Estimator** (Zhang et al., 2020) presents a KDE-based ECE estimator that relies on kernel density estimation to approximate the desired metric. Canonically, ECE is computed as an integral over the confidence space:

$$\hat{\text{ECE}}^d = \int \|z - \hat{\pi}(z)\|_d^d \hat{p}(z) dz \tag{9}$$

where $z = \{z_1, z_2, ..., z_L\}$ denotes model confidence distribution over $L$ classes, $\|\cdot\|_d^d$ denotes the $d^{th}$ power of the $\ell_d$ norm, and $\hat{p}(z)$ represents the marginal density function of model's confidence $\hat{\pi}(z)$ on a given dataset. The $\hat{p}(z)$ and $\hat{\pi}(z)$ are approximated using kernel density estimation. We argue that SECE is a special instance with $d = 1$ of $\hat{\text{ECE}}^d$ and estimate ECE with max probability $z_t, t = \arg\max\{z_1, z_2, ..., z_L\}$ for a single instance. And SECE is an upper bound of $\hat{\text{ECE}}^1$:

$$\hat{\text{ECE}} = \int |z - \hat{\pi}(z)| \hat{p}(z) dz \tag{10}$$

$$= \int |z_l - \hat{\pi}(z_l)| \hat{p}(z_l) dz_l \int |z_t - \hat{\pi}(z_t)| \hat{p}(z_t) dz_t \tag{11}$$

$$\leq \int |z_t - \hat{\pi}(z_t)| \hat{p}(z_t) dz_t = \text{SECE} \tag{12}$$

with $l = [1, L], l \neq t$ and density functions:

$$\hat{p}(z_t) = \frac{h^{-L}}{M} \sum_{i=1}^{M} K(z_t, z_i), \ \hat{\pi}_t(z_t) = \frac{\sum_{i=1}^{M} \pi(i) K(z_t, z_i)}{\sum_{i=1}^{M} K(z_t, z_i)} \tag{13}$$

where $h$ is the kernel width and $\pi(i)$ represents the binary accuracy of point $i$. The $\hat{p}(z_t)$ is a mixture of Dirac deltas centered on confidence predicted for individual points within the dataset (used to approximate ECE). Replacing binary accuracy with accuracy computed using an all-pairs approach (Qin et al., 2010) results in SECE, which is differentiable and can be computed efficiently for smaller batches of data since the integral is replaced with a sum over all examples in a batch.

**Discussion**: There exist other studies that develop a differentiable calibration error metrics, for instance, DECE (Bohdal et al., 2021) and SB-ECE (Karandikar et al., 2021). As pointed out in (Zhang et al., 2020), non-parametric density estimators are continuous, thus avoiding the binning step and potentially reduce binning biases. The difference between SECE and KDE-based estimators do not invalidate the analysis of the EĈE as an unbiased estimator of ECE (Theorem 4.1 in (Zhang et al., 2020)) as the all-pairs accuracy is an unbiased estimator of accuracy and the details of the $p(z)$ distribution are not used in the derivation beyond setting the bounds. As a result, the analysis described in (Zhang et al., 2020) to show their KDE-based approximation to ECE is unbiased can be re-used to show the same property for SECE .

### 4.3 Optimising $\gamma$-Net with SECE

The newly introduced $\gamma$-Net and the SECE metric can be applied together to optimize selected ML models for both cross-entropy and calibration error using the meta learning scheme introduced in Section 3.3. As the probability calibration via maximising entropy is performed in a continuous space, we argue that SECE is an efficient learning objective to optimize $\gamma$-Net and provide calibration regularization to base network trained with focal loss.

Under the umbrella of meta-learning, two sets of parameters are learned simultaneously: $\theta$, the parameters of base network $f^c$, as well as the $\phi$, the parameters of $\gamma$-Net denoted as $f^\gamma$. The former, $f^c$, is used to classify each new example into appropriate classes and the latter, $f^\gamma$, is applied to find the optimal value of the focal loss parameter $\gamma$ for each training example. Algorithm 1 in Appendix describes the learning procedures.

## 5 Experiments

We conducted a series of experiments to achieve the following objectives:

- Examine both the predictive and calibration performance of the proposed method. We used error (accuracy) to measure the predictive performance and Negative Log-Likelihood (NLL), Expected Calibration Error (ECE), Maximum Calibration Error (MCE), Adaptive Calibration Error (ACE), and Classwise ECE for calibration performance.

- Observe the calibration behaviours of the proposed method during training.

- Empirically evaluate the learned $\gamma$ values for Focal loss and the robustness of different methods to the binning mechanism.

**Implementation Details**. We implemented our methods by adapting and extending the code from (Bohdal et al., 2021) with Pytorch (Paszke et al., 2019). For all experiments we used their default settings (using ResNet18 as base model, batch size 128, data augmented with random crop and horizontal flip) unless otherwise stated. Each experiment was run 5 times with different random seeds, and results were averaged. We conducted our experiments on CIFAR-10 and CIFAR-100 (in (Bohdal et al., 2021)) as well as Tiny-ImageNet (Ya Le, 2015). For meta-learning, we split the training set into 8:1:1 as training/val/meta-validation, keeping the original test sets untouched. The experimental pipeline for all three datasets was

identical. The models were trained with SGD (learning rate 0.1, momentum 0.9, weight decay 0.0005) for up to 350 epochs. The learning rate was decreased at 150 and 250 epochs by a factor of 10. The model selection was based on the best validation error.

The $\gamma$-Net is implemented with a multi-head attention layer with $k$ heads and a fully-connected layer. $k$ is set to the number of categories. The hidden dimension is set to 512, the temperature $\tau$ is fixed at 0.01. For SECE, we used the Gaussian kernel with bandwidth of 0.01 (selected via grid search) for both datasets. We initialized $\gamma = 1.0$. During inference, the meta-network is not present excepted except in our ablation study on learned $\gamma$ values in Section 5.2.

**Baselines**. We extensively compare our method with baselines including standard cross-entropy (CE), cross-entropy with post-hoc temperature scaling (TS) (Platt et al., 1999), Focal loss with standard gamma value (Focal), $\gamma = 1$ (Lin et al., 2017), Focal Loss with scheduled gamma (FLSD) (Mukhoti et al., 2020), MMCE (Maximum Mean Calibration Error) (Kumar et al., 2018) and Label Smoothing with smooth factor 0.05 (LS-0.05) or (LS-0.1) (Müller et al., 2019) and Mix-Up ($\alpha = 1.0$) (Thulasidasan et al., 2019). In meta-learning setting, we include CE-DECE (meta-calibration(Bohdal et al., 2021) for learning unit-wise weight regularization), which serves as our meta-learning baseline, CE-SECE, FL$_\gamma$-DECE, focal loss with learnable sample-wise $\gamma$ and FL$_\gamma$-DECE.

Table 1: The predictive (test error) and calibration performance of different methods on CIFAR-10 (Top), CIFAR-100 (Middle) and Tiny-ImageNet (Bottom). The best scores are **bold**. The mean and standard deviation numbers are reported by averaging 5 runs with random seeds. As an alternative calibration method, our approach generally exhibits better calibration while retaining competitive predictive performance compared to conventional as well as meta-learning baselines.

| Methods | Error | NLL | ECE | MCE | ACE | Classwise ECE |
|---|---|---|---|---|---|---|
| | | | CIFAR 10 | | | |
| CE | $4.812 \pm 0.122$ | $0.335 \pm 0.01$ | $4.056 \pm 0.092$ | $33.932 \pm 5.433$ | $4.022 \pm 0.136$ | $0.848 \pm 0.023$ |
| CE (TS) | $4.812 \pm 0.122$ | $0.211 \pm 0.005$ | $3.083 \pm 0.140$ | $26.695 \pm 2.959$ | $3.046 \pm 0.157$ | $0.656 \pm 0.022$ |
| Focal | $4.874 \pm 0.100$ | $0.207 \pm 0.005$ | $3.193 \pm 0.104$ | $28.034 \pm 5.702$ | $3.174 \pm 0.098$ | $0.690 \pm 0.018$ |
| FLSD | $4.916 \pm 0.074$ | $0.211 \pm 0.005$ | $6.904 \pm 0.462$ | $\mathbf{19.246 \pm 11.071}$ | $6.805 \pm 0.446$ | $1.465 \pm 0.088$ |
| LS (0.05) | $4.744 \pm 0.126$ | $0.232 \pm 0.003$ | $2.900 \pm 0.085$ | $24.860 \pm 8.599$ | $3.985 \pm 0.154$ | $0.727 \pm 0.009$ |
| LS(0.1) | $4.918 \pm 0.085$ | $0.266 \pm 0.004$ | $7.566 \pm 0.41$ | $16.033 \pm 3.783$ | $7.611 \pm 0.161$ | $1.637 \pm 0.056$ |
| Mixup($\alpha$=1.0) | $\mathbf{4.126 \pm 0.068}$ | $0.273 \pm 0.033$ | $12.863 \pm 3.2$ | $20.739 \pm 4.205$ | $12.833 \pm 3.161$ | $2.678 \pm 0.615$ |
| MMCE | $4.808 \pm 0.082$ | $0.333 \pm 0.012$ | $4.027 \pm 0.082$ | $41.647 \pm 10.275$ | $4.013 \pm 0.091$ | $0.845 \pm 0.014$ |
| CE-DECE | $5.194 \pm 0.161$ | $0.301 \pm 0.038$ | $4.106 \pm 0.402$ | $41.346 \pm 13.325$ | $4.088 \pm 0.395$ | $0.868 \pm 0.074$ |
| CE-SECE | $5.222 \pm 0.168$ | $0.289 \pm 0.027$ | $4.062 \pm 0.241$ | $50.81 \pm 21.705$ | $4.049 \pm 0.251$ | $0.852 \pm 0.040$ |
| FL$_\gamma$-DECE | $5.434 \pm 0.095$ | $\mathbf{0.193 \pm 0.009}$ | $2.257 \pm 0.787$ | $56.633 \pm 23.856$ | $2.396 \pm 0.669$ | $\mathbf{0.557 \pm 0.165}$ |
| FL$_\gamma$-SECE | $5.428 \pm 0.144$ | $\mathbf{0.193 \pm 0.010}$ | $\mathbf{2.138 \pm 0.819}$ | $22.725 \pm 5.756$ | $\mathbf{2.357 \pm 0.541}$ | $\mathbf{0.556 \pm 0.165}$ |
| | | | CIFAR-100 | | | |
| CE | $22.570 \pm 0.438$ | $0.997 \pm 0.014$ | $8.380 \pm 0.336$ | $23.250 \pm 2.436$ | $8.347 \pm 0.344$ | $0.233 \pm 0.006$ |
| CE (TS) | $22.570 \pm 0.438$ | $0.959 \pm 0.008$ | $5.388 \pm 0.393$ | $13.454 \pm 2.377$ | $5.360 \pm 0.315$ | $0.208 \pm 0.003$ |
| Focal | $22.498 \pm 0.214$ | $0.900 \pm 0.007$ | $5.044 \pm 0.203$ | $12.454 \pm 0.893$ | $5.015 \pm 0.207$ | $0.203 \pm 0.004$ |
| FLSD | $22.656 \pm 0.113$ | $0.876 \pm 0.005$ | $5.956 \pm 0.804$ | $14.716 \pm 1.387$ | $5.958 \pm 0.802$ | $0.241 \pm 0.008$ |
| LS (0.05) | $21.810 \pm 0.172$ | $1.070 \pm 0.011$ | $8.108 \pm 0.346$ | $20.268 \pm 1.536$ | $8.106 \pm 0.346$ | $0.272 \pm 0.006$ |
| LS(0.1) | $22.244 \pm 0.155$ | $1.052 \pm 0.011$ | $4.754 \pm 0.709$ | $17.228 \pm 0.923$ | $4.777 \pm 0.647$ | $0.239 \pm 0.004$ |
| Mixup($\alpha$=1.0) | $\mathbf{21.210 \pm 0.227}$ | $0.917 \pm 0.017$ | $9.716 \pm 0.754$ | $16.01 \pm 1.335$ | $9.722 \pm 0.740$ | $0.315 \pm 0.011$ |
| MMCE | $22.490 \pm 0.143$ | $1.021 \pm 0.007$ | $8.713 \pm 0.245$ | $23.565 \pm 1.141$ | $8.670 \pm 0.305$ | $0.238 \pm 0.004$ |
| CE-DECE | $23.406 \pm 0.323$ | $1.148 \pm 0.006$ | $7.309 \pm 0.245$ | $22.565 \pm 1.446$ | $7.253 \pm 0.315$ | $0.241 \pm 0.002$ |
| CE-SECE | $23.448 \pm 0.302$ | $1.153 \pm 0.015$ | $7.668 \pm 0.330$ | $24.261 \pm 1.614$ | $7.609 \pm 0.295$ | $0.244 \pm 0.002$ |
| FL$_\gamma$-DECE | $23.712 \pm 0.204$ | $0.888 \pm 0.009$ | $\mathbf{1.879 \pm 0.440}$ | $8.271 \pm 2.651$ | $\mathbf{1.838 \pm 0.371}$ | $0.195 \pm 0.005$ |
| FL$_\gamma$-SECE | $23.686 \pm 0.377$ | $\mathbf{0.877 \pm 0.004}$ | $1.940 \pm 0.365$ | $\mathbf{7.480 \pm 1.867}$ | $1.939 \pm 0.379$ | $\mathbf{0.192 \pm 0.006}$ |
| | | | Tiny-ImageNet | | | |
| CE | $40.110 \pm 0.110$ | $1.838 \pm 0.171$ | $8.059 \pm 1.296$ | $15.73 \pm 1.905$ | $8.006 \pm 1.282$ | $0.154 \pm 0.001$ |
| Focal | $39.415 \pm 0.625$ | $1.896 \pm 0.009$ | $7.600 \pm 0.309$ | $13.771 \pm 0.897$ | $7.469 \pm 0.301$ | $0.152 \pm 0.002$ |
| FLSD | $39.705 \pm 0.075$ | $1.904 \pm 0.025$ | $14.501 \pm 1.078$ | $21.528 \pm 2.116$ | $14.501 \pm 1.078$ | $0.202 \pm 0.006$ |
| LS (0.1) | $\mathbf{39.395 \pm 0.305}$ | $2.185 \pm 0.001$ | $16.777 \pm 0.476$ | $29.088 \pm 1.835$ | $16.901 \pm 0.460$ | $0.199 \pm 0.001$ |
| Mixup($\alpha$=1.0) | $39.890 \pm 0.271$ | $1.932 \pm 0.054$ | $12.133 \pm 2.069$ | $31.440 \pm 0.968$ | $12.028 \pm 2.079$ | $0.193 \pm 0.009$ |
| MMCE | $40.310 \pm 0.100$ | $1.826 \pm 0.177$ | $8.206 \pm 1.219$ | $16.802 \pm 2.339$ | $8.165 \pm 1.269$ | $\mathbf{0.149 \pm 0.001}$ |
| CE-DECE | $41.350 \pm 0.000$ | $2.228 \pm 0.033$ | $10.694 \pm 0.503$ | $20.888 \pm 0.430$ | $10.553 \pm 0.553$ | $0.160 \pm 0.000$ |
| CE-SECE | $41.005 \pm 0.145$ | $2.213 \pm 0.058$ | $10.928 \pm 1.125$ | $21.362 \pm 2.526$ | $10.912 \pm 1.069$ | $0.157 \pm 0.003$ |
| FL$_\gamma$-DECE | $40.625 \pm 0.095$ | $\mathbf{1.826 \pm 0.007}$ | $5.944 \pm 1.090$ | $11.542 \pm 1.990$ | $6.077 \pm 1.095$ | $0.155 \pm 0.007$ |
| FL$_\gamma$-SECE | $40.850 \pm 0.140$ | $1.829 \pm 0.005$ | $\mathbf{5.794 \pm 0.756}$ | $\mathbf{11.477 \pm 1.563}$ | $\mathbf{5.848 \pm 0.751}$ | $0.156 \pm 0.005$ |

### 5.1 Predictive and Calibration Performance

Table 1 presents the performance comparison across approaches. Temperature Scaling (TS) can effectively reduce the errors in calibration metrics compared to uncalibrated CE models (baseline). Label smoothing and Mixup achieve the best test errors on datasets but exhibit higher ECE and MCE scores. Focal loss and FLSD can improve calibration in general, but we also observed high MCE score for FLSD. On the other hand, we found that MMCE exhibits higher calibration errors as compared to baseline, this aligns with the findings from (Bohdal et al., 2021). We further discussed the predictive-calibration trade-off in Appendix A.4. Our proposed approach $\text{FL}_\gamma$-SECE achieves lower errors on most of calibration metrics with competitive predictive performance. Compared to the prior work (CE-DECE), our method ($\text{FL}_\gamma$-SECE) showed significant calibration improvement across multiple calibration metrics. Particularly, on MCE (which measures the worst-case mismatch between accuracy and confidence), there are 18.62%, 15.09% and 9.41% improvements on CIFAR-10, CIFAR-100 and Tiny-ImageNet, respectively.

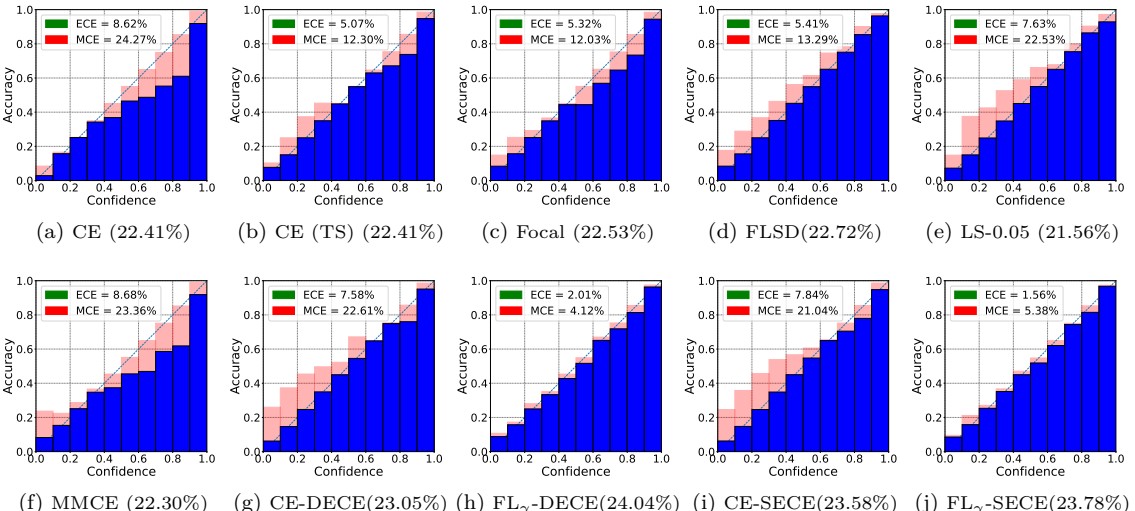

Figure 2: The reliability diagram plot for models on CIFAR-100 test set. The ($\cdot$) represents test error. The diagonal dash line represents perfect calibration. The red bar represents the gap between the observed accuracy and the desired accuracy of the perfectly calibrated model (the diagonal) - it is positive if the observed accuracy is lower and negative otherwise. The model from the $5^{th}$ run is used.

Figure 2 illustrates the reliability diagram plots of models on CIFAR-100. (The plots for CIFAR-10 are in Appendix Figure 7).

The diagrams show that for CIFAR-100, $\gamma$-Net -based methods achieve better ECE compared to SECE and meta-loss for optimizing $\gamma$-Net ensures smooth and stable calibrations, potentially reduces calibration biases in bins. Among compared methods, our method $\text{FL}_\gamma$-SECE achieves considerably lower errors on calibration metrics. When comparing meta-learning based approaches: CE-DECE (meta-learning baseline), CE-SECE, $\text{FL}_\gamma$-DECE, $\text{FL}_\gamma$-SECE. We can see that our method ($\text{FL}_\gamma$-SECE) achieves comparable test error and better scores across calibration metrics. Particularly, it improves ECE by an average of roughly 4.198% on three datasets, additionally improving MCE by an average of 14.37% MCE and ACE score by 3.917%.

To observe the learning behavior of models, we plotted the curves showing test ECE changes in Figure 3 (a-b), and test error (included in Appendix Section A.3.1). It is noteworthy that while the test error curves exhibit similar behaviors, the calibration behaviors vary significantly across methods. The primary advantage of $\gamma$-Net with SECE lies in its stable and smooth calibration behavior. After the $150^{th}$ epoch, when the learning rate is decreased by a factor of 10, we observe an increase in ECE scores for all approaches, particularly for FLSD (green lines). This indicates that during model training, the learning procedure aims to align the probability distribution as closely as possible with the ground-truth distribution (one-hot representation),

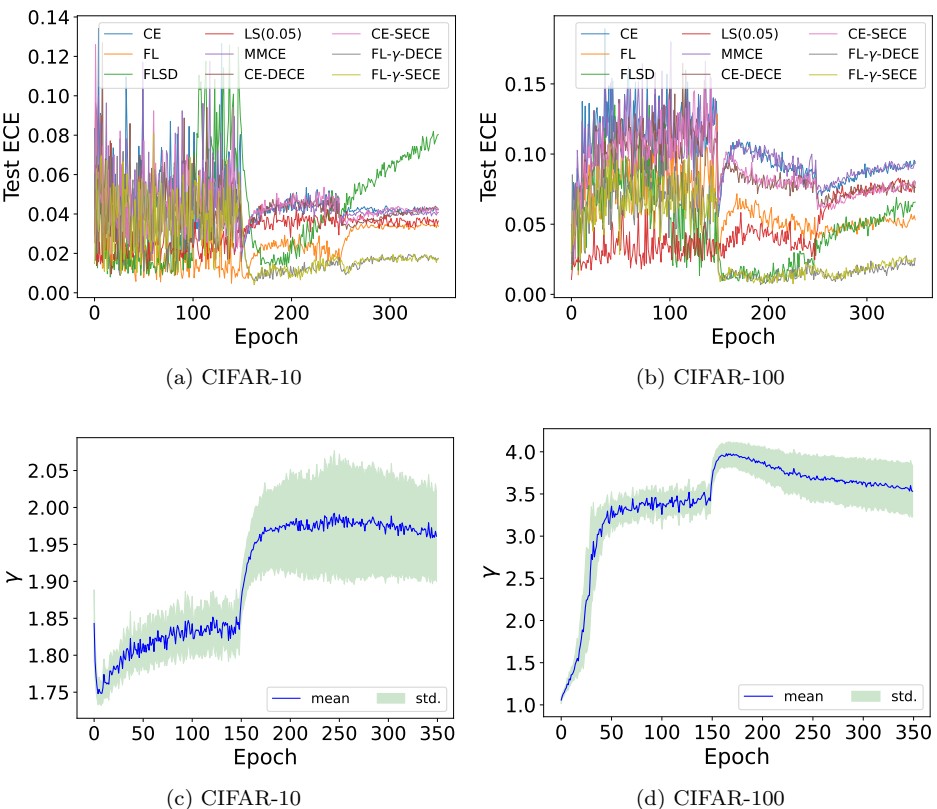

Figure 3: (a-b): ECE curves on the test dataset of CIFAR-10 (a) and CIFAR-100 (b).(c-d): The mean and standard deviation (std.) of $\gamma$ on test dataset at each epoch. Low std. score indicates samples share similar gamma values, and high std. score indicates more samples have different $\gamma$ values.

causing models to become overconfident. It also demonstrates that our approach is more robust to this phenomenon.

## 5.2 Ablation Study

### 5.2.1 Learning $\gamma$ as continous variables

Figure 3(c-d) presents the changes in $\gamma$ values on test dataset over epochs. In the earlier stages of training, the sample-wise $\gamma_j$ for $x_j \in D_{test}$ have similar values (with low variance) because they are initialized with $\gamma_i = 1.0$ for $x_i \in D_{val}$. The $\gamma$ parameter is observed to have a higher standard deviation in the later stage of training as $\gamma$-Net learns the optimal value for each example in the dataset rather than relying on global values, showcasing the flexibility of the network. It is also noted that $\gamma$ is learned in a continuous space, which is different to the discrete values the pre-defined in Focal Loss (Lin et al., 2017) and FLSD (Mukhoti et al., 2020).

### 5.2.2 Calibration bias and robustness

In Figure 5, we examine the robustness of those methods with different binning schemes by varying the number of bins, which is one of causes of introducing calibration bias (Minderer et al., 2021). It shows that $\gamma$-Net based approaches (FL$_\gamma$-DECE and FL$_\gamma$-SECE) maintain much lower ECE score when throughout all bin numbers from 10 to 1000 showing the trained based network is robustly calibrated. Furthermore FL$_\gamma$-SECE is also able to maintain lower MCE as compared to other methods.

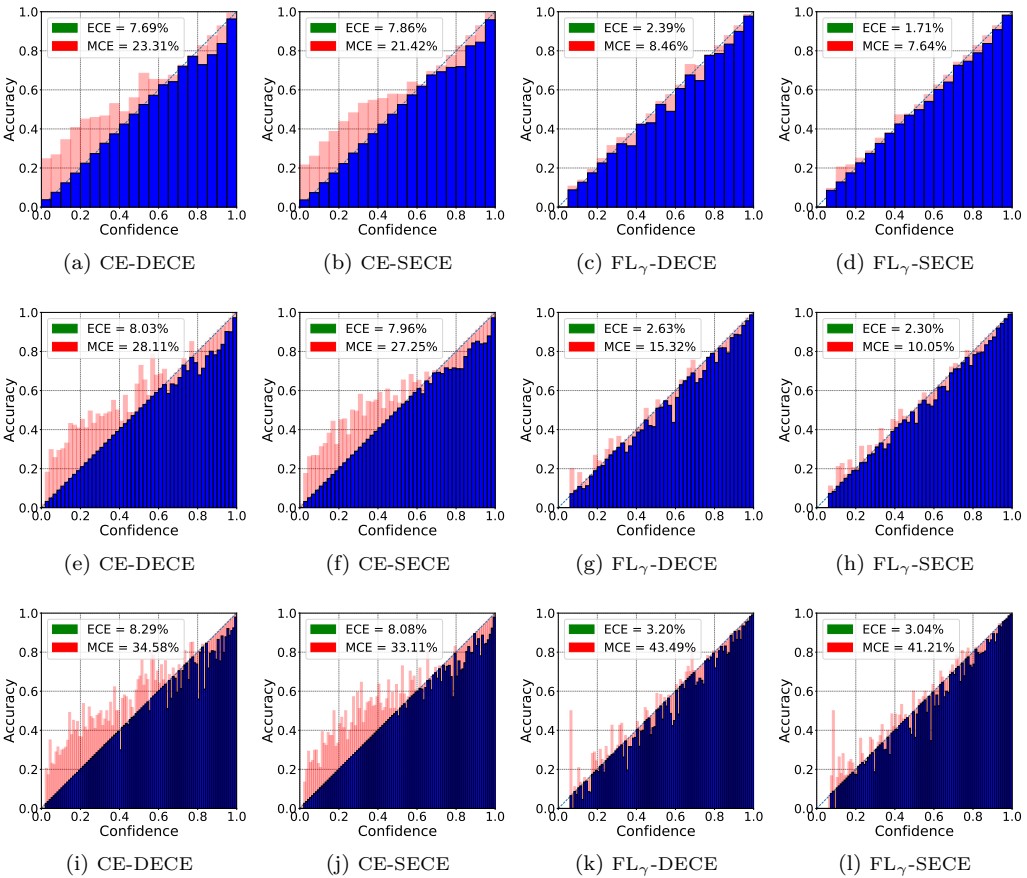

Figure 4: The reliability diagram plots on CIFAR-100 with large bin numbers (top to bottom: 20, 50, 100).

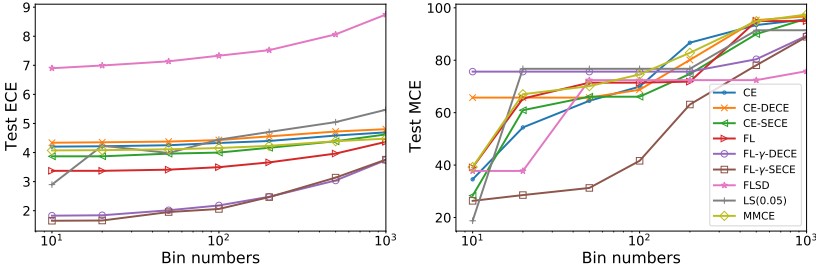

Figure 5: The changes in ECE (left) and MCE (right) scores on the CIFAR-10 test dataset with increasing bin numbers in the range of [10, 20, 50, 100, 200, 500, 1000] are illustrated. $FL_\gamma$-SECE demonstrates superior robustness to increasing bin numbers, as evidenced by lower MCE. A similar plot for CIFAR-100 is provided in the Figure 8 of the Appendix.

Here we can clearly see the role of SECE in stabilizing the calibration with $\gamma$-Net against the change in the number of bins. Table 2 shows the individual calibration gain from SECE as compared to using learnable $\gamma$ only with our method $FL_\gamma$-SECE. With SECE, we can see improved predictive and calibration performance. Importantly, from Figure 4 and Figure5, we observe that it reduces calibration bias introduced by different binning mechanisms. This is mostly reflected in reduced ECE/MCE scores, i.e., the model exhibits high MCE on a particular bin if calibration bias exists.

| Methods | Error | NLL | ECE | MCE | ACE | Classwise ECE |
|---------|-------|-----|-----|-----|-----|---------------|
| FL$_\gamma$ | $5.632 \pm 0.118$ | $0.197 \pm 0.009$ | $2.177 \pm 0.619$ | $46.172 \pm 28.240$ | $\mathbf{2.319 \pm 0.407}$ | $\mathbf{0.553 \pm 0.120}$ |
| FL$_\gamma$-SECE | $\mathbf{5.428 \pm 0.144}$ | $\mathbf{0.193 \pm 0.010}$ | $\mathbf{2.138 \pm 0.819}$ | $\mathbf{22.725 \pm 5.756}$ | $2.357 \pm 0.541$ | $0.556 \pm 0.165$ |
| FL$_\gamma$ | $28.148 \pm 8.127$ | $1.051 \pm 0.278$ | $3.044 \pm 1.542$ | $10.082 \pm 3.441$ | $3.016 \pm 1.511$ | $0.226 \pm 0.063$ |
| FL$_\gamma$-SECE | $\mathbf{23.686 \pm 0.377}$ | $\mathbf{0.877 \pm 0.004}$ | $\mathbf{1.940 \pm 0.365}$ | $\mathbf{7.480 \pm 1.867}$ | $\mathbf{1.939 \pm 0.379}$ | $\mathbf{0.192 \pm 0.006}$ |

Table 2: The calibration gain from SECE on CIFAR-10 (top) and CIFAR-100 (bottom).

## 6 Discussion

**Novelties**. Though we follow the general setup in (Bohdal et al., 2021), our work makes several novel contributions: (1) We introduce learnable sample-wise continuous variables for focal loss. To the best of our knowledge, this is the first work to introduce fine-grained $\gamma$ for focal loss for the purpose of model calibration, learned from a meta-network. (2) We propose SECE, which reduces model calibration bias across different binning mechanisms, an aspect rarely discussed in the literature. Additionally, SECE is also more efficient than DECE in (Bohdal et al., 2021) as it uses simple summation rather than predicting the bin assignment using networks. (3) We further showcase the importance and feasibility of using gradient-based meta-learning (Finn et al., 2017) in alleviating model miscalibration. As shown in Table 3, meta-regularization can achieve very competitive predictive and calibration performance compared to conventional regularization methods.

| Methods | ECE(%) | Test Error(%) |
|---------|--------|---------------|
| CIFAR-10 | | |
| (Patra et al., 2023) | 0.59 | 6.28 |
| (Hebbalaguppe et al., 2022) | 0.93 | 7.18 |
| (Hebbalaguppe et al., 2022) | 0.70 | 7.08 |
| (Liu et al., 2022) (without marign) | 3.72 | 5.24 |
| (Liu et al., 2022) | 1.16 | 4.75 |
| (Karandikar et al., 2021) (Focal+SB-ECE) | 1.19 | 4.90 |
| (Karandikar et al., 2021) (Focal+SB-AvUC ) | 1.58 | 5.60 |
| (Tao et al., 2023) (Dual Focal with ResNet50) | 0.46 | 5.17 |
| (Tao et al., 2023) (Dual Focal with ResNet110) | 0.98 | 5.02 |
| (Ghosh et al., 2022) (AdaFocal with ResNet50) | 0.66 | 5.30 |
| (Ghosh et al., 2022) (AdaFocal with ResNet50) | 0.71 | 5.27 |
| Meta-Regularization (Ours) | 2.14 | 5.43 |
| CIFAR-100 | | |
| (Patra et al., 2023) | 1.74 | 26.57 |
| (Hebbalaguppe et al., 2022) | 1.49 | 31.58 |
| (Hebbalaguppe et al., 2022) | 0.72 | 29.80 |
| (Karandikar et al., 2021)(Focal+SB-ECE) | 2.30 | 21.40 |
| (Karandikar et al., 2021) (Focal+SB-AvUC ) | 1.57 | 21.90 |
| (Tao et al., 2023) (Dual Focal with ResNet50) | 1.08 | 22.67 |
| (Tao et al., 2023) (Dual Focal with ResNet110) | 2.90 | 22.59 |
| (Ghosh et al., 2022) (AdaFocal with ResNet50) | 1.36 | 22.60 |
| (Ghosh et al., 2022) (AdaFocal with ResNet50) | 1.40 | 22.79 |
| Meta-Regularization (Ours) | 1.94 | 23.69 |

Table 3: Comparing meta-regularization (ours) to recent (non-meta) regularization methods on CIFAR-10 and CIFAR-100. Meta-regularization provides very competitive predictive and calibration performance compared to conventional regularization methods (the numbers are obtained from the corresponding papers).

**Limitations**. Our proposed method utilizes $\gamma$-Net for learning $\gamma$, which increases the number of parameters by approximately 0.46% during training. However, since $\gamma$-Net is not retained after training, there is no additional computational complexity during inference.

## 7 Conclusion

In this work, we have introduced a meta-learning based approach for acquiring well-calibrated models and demonstrated the advantages of two newly introduced components. By learning a sample-wise $\gamma$ for Focal

loss using $\gamma$-Net, we achieve both strong predictive performance and unbiased, robust calibration. The optimization of $\gamma$-Net with SECE proves crucial in ensuring stable calibration compared to baselines. Through extensive empirical results on three computer vision datasets, we have demonstrated that our method enhances calibration capability without altering the original networks.

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

# A  Appendix

## A.1  Evaluation Metrics

Besides ECE and MCE, we evaluate models on Classwise ECE and Adaptive ECE.

**Classwise ECE** (Nixon et al., 2019). Classwise ECE extends the bin-based ECE to measure calibration across all the possible classes. In practice, predictions are binned separately for each class and the calibration error is computed at the level of individual class-bins and then averaged. The metric can be formulated as

$$\text{CECE} = \sum_{m=1}^{M} \sum_{c=1}^{K} \frac{|b_{m,c}|}{NK} |\text{ACC}(b_{m,c}) - \text{CONF}(b_{m,c})| \tag{14}$$

where $N$ is the total number of samples, $K$ is the number of classes and $b_{m,c}$ represents a single bin for class $c$. In this formulation, $\text{ACC}(b_{m,c})$ represents average binary accuracy for class $c$ over bin $b_{m,c}$ and $\text{CONF}(b_{m,c})$ represents average confidence for class $c$ over bin $b_{m,c}$.

**Adaptive ECE** (Nixon et al., 2019). Adaptive ECE further extends the Classwise variant of expected calibration error by introducing a new binning strategy which focuses the measurement on the confidence regions with multiple predictions and. Concretely Adaptive ECE (ACE) spaces the bin intervals such that each contains an equal number of predictions. The metric can be formulated as

$$\text{ACE} = \sum_{r=1}^{R} \sum_{c=1}^{K} \frac{1}{RK} |\text{ACC}(b_{r,c}) - \text{CONF}(b_{r,c})| \tag{15}$$

where $r$ is a calibration range that is defined by the $[\frac{N}{R}]$-th index of the sorted and thresholded predictions. In this formulation, $\text{ACC}(b_{r,c})$ represents the average accuracy for class $c$ over the calibration range $r$ and $\text{CONF}(b_{r,c})$ represents the average confidence for class $c$ over calibration range $r$.

## A.2  Algorithm

Algorithm 1 describes the learning procedures, it relies on the training and validation sets $D_{train}$, $D_{val}$, two networks $f^c$ and $f^\gamma$ with parameters $\theta$ and $\phi$. The optimization proceeds in an iterative process until both sets converge. Each iteration starts by sampling a mini-batch of training data $(\mathbf{x}_i^t, \mathbf{y}_i^t) \sim D_{train}$ as well as a mini-batch of validation data $(\mathbf{x}_i^v, \mathbf{y}_i^v) \sim D_{val}$. The training mini-batch is used to find the optimal value of $\gamma$ as $\gamma = f^\gamma(\mathbf{x}_i^t)$ and following that, compute the focal loss on the outputs of the backbone network $f^c$. The gradient of the loss $\mathcal{L}_\gamma^f(f^c(\mathbf{x}_i^t), \mathbf{y}_i^t)$ is then used to update the parameters of the backbone network $\theta$ using the Adam algorithm (Kingma & Ba, 2014).

Once the backbone model $f^c$ is updated, it is used to compute the value of the auxiliary loss SECE on the validation mini-batch SECE $(f^c(\mathbf{x}_i^v), \mathbf{y}_i^v)$. The auxiliary SECE loss is a differentiable proxy of the true expected calibration error and can be minimized w.r.t. $\phi$ to find the optimal parameters of $\gamma$-Net. It is important to note that SECE is computed based on the outputs of $f^c(\mathbf{x}_i^v)$ and the only dependence on $\gamma$ is an indirect one based on the updates of $\theta$, as discussed in Section 3.3.

## A.3  Additional Experiments

### A.3.1  Test Error Curves

Figure 6 presents the test error on CIFAR-10 and CIFAR-100 for the compared methods. In general, they exhibit similar behavior.

### A.3.2  Reliability Diagram Plots

Figure 8 examines the robustness of these methods with different binning schemes by varying the number of bins. It shows that $\gamma$-Net based approaches (FL$_\gamma$-DECE and FL$_\gamma$-SECE) maintain much lower ECE score throughout all bin numbers from 10 to 1000 indicating that the trained based network is robustly calibrated.

---

**Algorithm 1:** Meta optimization with $\gamma$-Net and SECE

---

**Input:** $f^c$ and $f^\gamma$ with initialized $\theta$ and $\phi$

**Output:** Optimized $\theta$ and $\phi$

**Data:** Training and validation sets: $D_{train}, D_{val}$

**1 while** $\theta$ *not converged* **do**

**2**     $(\mathbf{x}_i^t, \mathbf{y}_i^t) \sim D_{train}$;     $(\mathbf{x}_i^v, \mathbf{y}_i^v) \sim D_{val}$; # Sample a mini-batch from both datasets

**3**     $\gamma_i = f^\gamma(\mathbf{x}_i^t)$; # learning sample-wise $\gamma_i$

**4**     $\mathcal{L}_\gamma^f = \mathcal{L}_\gamma^f(f^c(\mathbf{x}_i^t), \mathbf{y}_i^t)$: # Compute training loss based on $\gamma$ $\theta := \theta - \eta\nabla_\theta \mathcal{L}_\gamma^f$: # Update parameters of $f^c$ using training loss gradient

**5**     $SECE = SECE\,(f^c(\mathbf{x}_i^v), \mathbf{y}_i^v)$ # Compute SECE auxiliary loss using validation batch

**6**     $\phi := \phi - \eta_\phi \nabla_\phi SECE$ : # Update parameters of $f^\gamma$ using SECE gradient

---

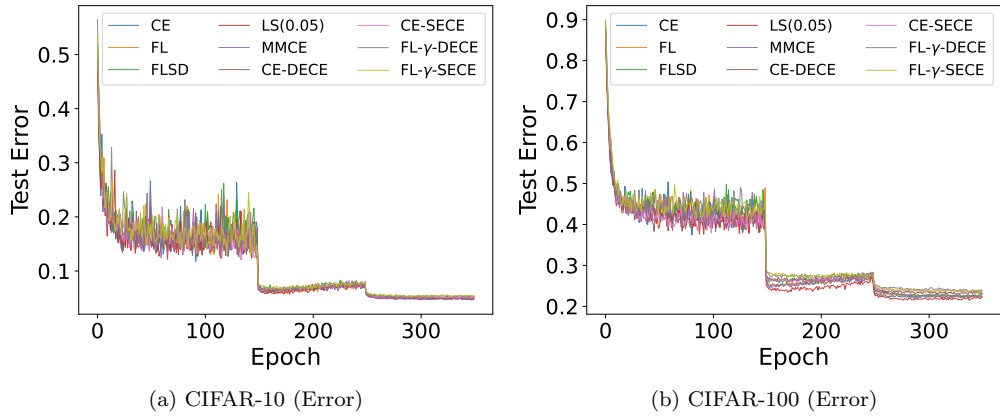

(a) CIFAR-10 (Error)         (b) CIFAR-100 (Error)

Figure 6: Test error on both test datasets.

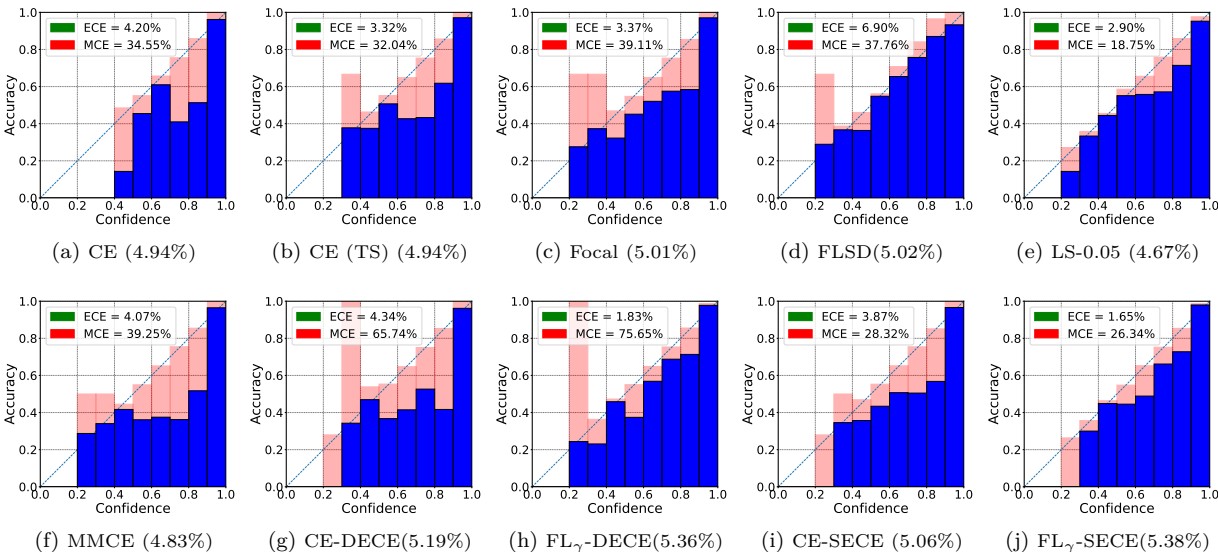

(a) CE (4.94%)    (b) CE (TS) (4.94%)    (c) Focal (5.01%)    (d) FLSD(5.02%)    (e) LS-0.05 (4.67%)

(f) MMCE (4.83%)    (g) CE-DECE(5.19%)    (h) FL$_\gamma$-DECE(5.36%)    (i) CE-SECE (5.06%)    (j) FL$_\gamma$-SECE(5.38%)

Figure 7: The reliability diagram plots for models on CIFAR-10 test set. The $(\cdot)$ denotes test error. The diagonal dashed line represents perfect calibration. The red bar represents the gap between the observed accuracy and the desired accuracy of the perfectly calibrated model (diagonal) - it is positive if the observed accuracy is lower and negative otherwise. The model from the $5^{th}$ run is used.

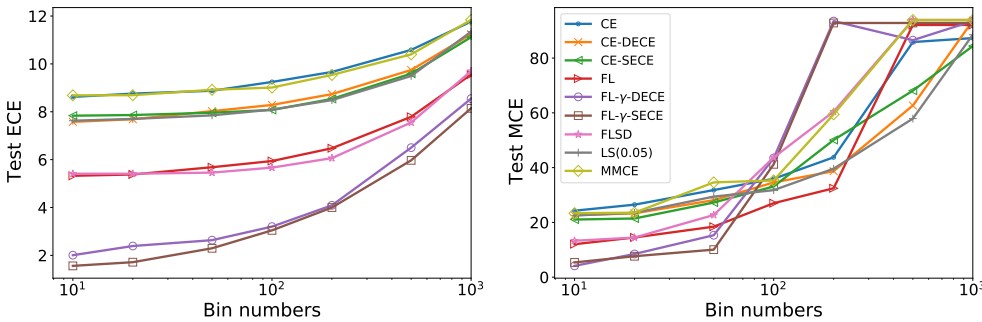

Figure 8: The changes in ECE (left) and MCE (right) scores on the CIFAR-100 test dataset with increasing bin numbers in the range of [10, 20, 50, 100, 200, 500, 1000] are illustrated. Our proposed approach shows better robustness on bin sizes.

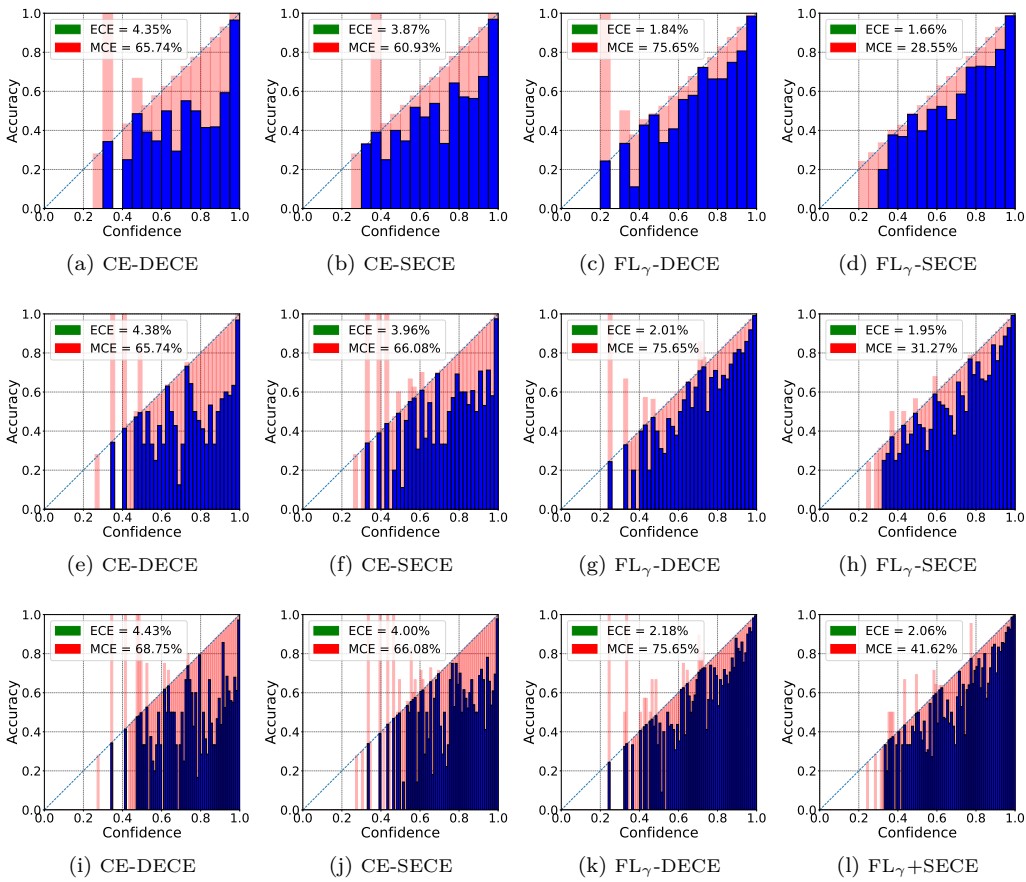

Figure 9: The reliability diagram plots for models on CIFAR-10 with large bin numbers (from top to bottom: 20, 50, 100). The diagonal dashed line represents perfect calibration, while red bar indicates the deviation from perfect calibration in each bin. The model from the $5^{th}$ run is used.

In Figure 9, we provide the comparison across meta-learning based approaches via reliability plots with large bin numbers and corresponding ECE and MCE. $FL_\gamma$-SECE is able to maintain the lowest ECE and MCE; it shows robustness to increased bin numbers.

| Methods | Error | NLL | ECE | MCE | ACE | Classwise ECE |
|---|---|---|---|---|---|---|
| CE | $5.354 \pm 0.097$ | $0.220 \pm 0.015$ | $3.038 \pm 0.159$ | $33.419 \pm 19.597$ | $3.035 \pm 0.157$ | $0.677 \pm 0.031$ |
| CE-DECE | $5.850 \pm 0.421$ | $0.236 \pm 0.017$ | $3.245 \pm 0.471$ | $25.062 \pm 2.184$ | $3.240 \pm 0.471$ | $0.723 \pm 0.072$ |
| CE-SECE | $5.895 \pm 0.271$ | $0.246 \pm 0.032$ | $3.555 \pm 0.428$ | $24.518 \pm 3.613$ | $3.555 \pm 0.425$ | $0.756 \pm 0.083$ |
| $FL_\gamma$-DECE | $6.084 \pm 0.188$ | $0.199 \pm 0.012$ | $2.151 \pm 1.499$ | $24.644 \pm 11.227$ | $2.139 \pm 1.457$ | $0.605 \pm 0.201$ |
| $FL_\gamma$-SECE | $6.263 \pm 0.266$ | $0.208 \pm 0.021$ | $2.549 \pm 2.233$ | $20.220 \pm 5.235$ | $2.559 \pm 2.217$ | $0.672 \pm 0.355$ |

Table 4: DenseNet (100 layers) on CIFAR-10 with meta-learning-based methods, 5 repetitions. gamma-net based methods exhibit better calibration across multiple metrics (NLL, ECE, ACE, Classwise ECE) while maintaining competitive test error.

### A.3.3 Experiments with DenseNet

We conducted experiments on DenseNet with meta-learning baselines. Table 4 presents the effectiveness of gamma-net based methods, which exhibit better calibration across multiple metrics with comparable predictive performance.

### A.4 The Predictive-Calibration Trade-off

Figure 10 depicts the empirical trade-off between predictive (measured by test error) and calibration (measured by ECE) performance. We can observe that many methods exist on the Pareto front. For instance, mixup ($\alpha = 0.1$), LS(0.1) and $FL_\gamma$-SECE achieves the low test error but high ECE on CIFAR datasets. This indicates there is predictive-calibration trade-off for most of methods. Similar Pareto fronts are observed in pretained large language models (LLMs) (Stengel-Eskin & Van Durme, 2023). However, as shown in Table 5, when evaluating the ECE improvement for our proposed $FL_\gamma$-SECE method, the reduction in predictive capability is modest.

| Methods | CIFAR-100 | | CIFAR-100 | | Tiny-ImageNet | |
|---|---|---|---|---|---|---|
| | $(\Delta_{error})$ | $\Delta_{ECE}$ | $(\Delta_{error})$ | $\Delta_{ECE}$ | $(\Delta_{error})$ | $\Delta_{ECE}$ |
| Focal | 0.062 | -0.863 | -0.072 | -3.336 | -0.695 | -0.459 |
| FLSD | 0.104 | 2.848 | 0.086 | -2.424 | -0.405 | 6.442 |
| LS(1.0) | 0.106 | 3.510 | -0.326 | -3.626 | -0.715 | 8.718 |
| Mixup($\alpha = 1.0$) | -0.686 | 8.807 | -1.360 | 1.336 | -0.220 | 4.074 |
| MMCE | -0.004 | -0.029 | -0.080 | 0.333 | 0.200 | 0.147 |
| CE-DECE | 0.382 | 0.050 | -0.164 | -1.071 | 1.240 | 2.635 |
| $FL_\gamma$-SECE | 0.616 | -1.918 | 1.116 | -6.440 | 0.740 | -2.265 |

Table 5: The test error difference ($\Delta_{error}$) and ECE difference ($\Delta_{ECE}$) for different methods compared to the baseline (CE).

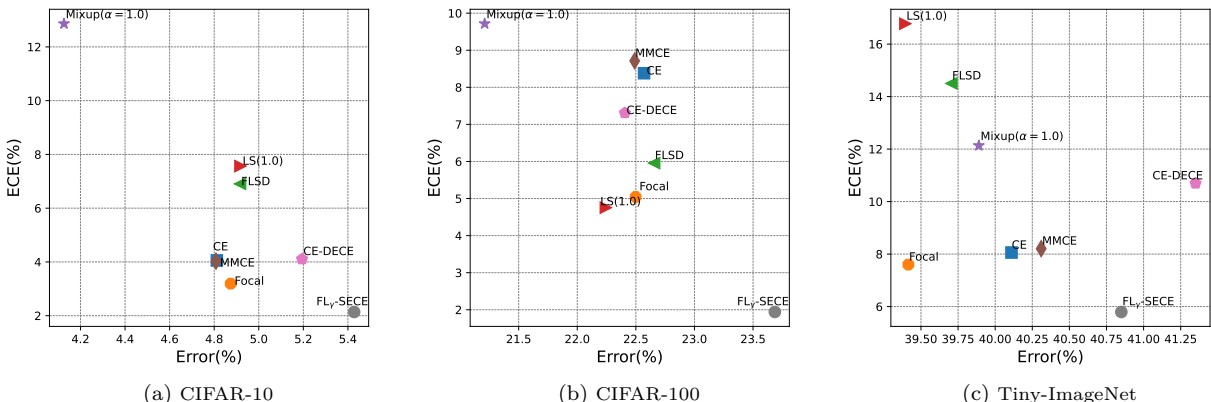

(a) CIFAR-10          (b) CIFAR-100          (c) Tiny-ImageNet

Figure 10: The trade-off between predictive and calibration performance of different methods on the test set of CIFAR-10, CIFAR-100 and Tiny-ImageNet. The Pareto front are observed for most of methods.

