# OpenReview forum: "Towards Unbiased Calibration using Meta-Regularization"
_TMLR — Accepted by TMLR_

### Review · Reviewer_fXcz · 2024-04-09

**Summary Of Contributions:**

This paper addresses the problem of calibrating neural networks. The introduction assumes that neural networks are ill-calibrated and proposes a new model and new loss function to train better-calibrated networks. Experimental results show that the method of sample-wise prediction of focal loss during training can improve the ECE values. However, both Table 1 and Figure 5 show no significant difference between DECE (prior work) and the proposed method. Therefore, the second claimed contribution remains to be argued about.

**Audience:**

Yes

**Broader Impact Concerns:**

n.a. better calibration would improve many applications.

**Claims And Evidence:**

No

**Requested Changes:**

* The definition of pi(x) is not clearly defined. The text states 'and pi(xi) are the confidence and accuracy of ith example.', but what does that mean? do you mean it's the **difference** between confidence and accuracy?
* Figure 3a: The ECE increases during training for most of the methods. This effect also seems present for the proposed method, although to a lesser extent. The text makes this observation, but no explanation is given. How could the ECE significantly increase when the learning is a proper scoring rule?
* Figure 5: there seems to be no significant difference between DECE and SECE. Could one conclude that most improvement come from the new focal loss and not the new Differentiable loss function?

Typo's?

 * 'Learnable sample-wise at as continuous variables for focal loss.'
 * 'Additionally, SECE is more also efficient than DECE''
 * 'We further showcase the importance and fesiablity'

**Strengths And Weaknesses:**

Strengths:

* Strengths: the proposed loss function is differentiable w.r.t. to both the model parameters and the focal loss parameter. The experimental results show that the proposed method can improve the calibration of the model by taking steps on the gradient of this loss function.
* Experimental results compare multiple methods across three datasets. Improvements are shown in the ECE values, which is a common metric for calibration. The results are consistent across datasets, which is a good sign that the method does not overfit a specific dataset.

Weaknesses:

*  The motivation for 'better calibration' is not clear. In most experimental results, the classwise calibration error is an order of magnitude lower than the error rate. In the case of CIFAR100, the classwise ECE is even two orders of magnitude lower than the error rate. What does an improvement in ECE then mean?
* Following on the previous question, most improvements are well within the noise standard deviation of DECE, which is prior work. Therefore, are the results significant?

---

> ### Author Response · Authors · 2024-04-24
> **Author Response to Reviewer fXcz**
>
> We truly appreciate the reviewer for their valuable comments. We address the concerns of the review below.
>
> >The motivation for 'better calibration' is not clear. In most experimental results, the classwise calibration error is an order of magnitude lower than the error rate. In the case of CIFAR100, the classwise ECE is even two orders of magnitude lower than the error rate. What does an improvement in ECE then mean?
>
> Thank you for raising this concern. Error rate and classwise ECE measure different aspects of model capability and as a result, they are not directly comparable. The former assesses predictive performance while the latter evaluates the calibration performance; therefore, they should be compared separately. Results in Table 1 shows that our method achieves mostly better classwise ECE (the lowest classwise ECE on CIFAR10 and CIFAR100 datasets), for instance, FLSD achieves 1.465 and our approach (FL$_{\gamma}$-SECE) achieves 0.556 which is a statistically significant improvement. We have further clarified the used metrics at the beginning of Experiment Section.
>
> >Following on the previous question, most improvements are well within the noise standard deviation of DECE, which is prior work. Therefore, are the results significant?
>
> > Figure 5: there seems to be no significant difference between DECE and SECE. Could one conclude that most improvement come from the new focal loss and not the new Differentiable loss function?
>
> As compared to the prior work (CE-DECE), our method ($FL_\gamma$-SECE) showed significant calibration improvement across multiple calibration metrics. The reported values are more than one standard deviation apart. Particularly, on MCE (which measures the worst-case mismatch between accuracy and confidence), there are 18.62\%, 15.09\% and 9.41\% improvements on CIFAR10, CIFAR100 and Tiny-ImageNet respectively. The difference between $FL_{\gamma}$-SECE and $FL_{\gamma}$-DECE are indeed within one standard deviation apart in most cases but we attribute it to the overall low values of calibration errors of $FL_{\gamma}$-X calibration methods meaning the difference is difficult to measure; still the average calibration error for $FL_{\gamma}$-SECE is consistently lower. Additionally, Figure 5(right) shows that $FL_{\gamma}$-SECE is more robust to increased bin numbers by exhibiting lower MCE score as compared to $FL_{\gamma}$-DECE. We added this to Figure 5 caption and make Figure 5 to be more self-contained.
>
> > The definition of pi(x) is not clearly defined. The text states 'and pi(xi) are the confidence and accuracy of ith example.', but what does that mean?
>
> The $z_i$ and $\pi(x_i)$ respectively denote the confidence and accuracy of the $i^{th}$ example, we have revised this to avoid confusion.
>
> > Figure 3a: The ECE increases during training for most of the methods. This effect also seems present for the proposed method, although to a lesser extent. The text makes this observation, but no explanation is given. How could the ECE significantly increase when the learning is a proper scoring rule?
>
> The increase in ECE is mostly due to the learning procedure attempting to push the predicted probability distribution to be as close as possible to the ground-truth distribution [1], which is usually a one-hot distribution for classification tasks. Without appropriate regularization, miscalibration (primarily overconfidence) occurs, leading to an increase in the ECE score. This demonstrates that our approach suffers less from this phenomenon.
>
> [1] Mukhoti et al., Calibrating Deep Neural Networks using Focal Loss. NeurIPS 2020.
>
> > Typo's?
>
> Thank you so much for your careful reading, we have fixed those typos.
>
> Once again, we greatly appreciate your diligent efforts in reviewing our paper. Please let us know if you have further concerns.

---

### Review · Reviewer_mW6C · 2024-04-13

**Summary Of Contributions:**

The paper proposes a meta-learning approach to address model miscalibration in deep neural networks. It introduces two main components: the Gamma-Net, which outputs sample-wise gamma values for focal loss, enabling fine-grained calibration regularization, and the Smooth Expected Calibration Error (SECE), an unbiased and differentiable calibration error metric using a Gaussian kernel. This combination allows for effective, sample-wise calibration adjustment. Tested across three computer vision datasets, the approach significantly enhances calibration performance without sacrificing predictive accuracy, showcasing its potential for practical deployment in various applications.

**Audience:**

Yes

**Broader Impact Concerns:**

The paper proposes a meta-learning approach to address model miscalibration in deep neural networks. Gamma-Net is an adaptive way to apply focal loss. The adaptive way can also impact other methods of this field and bring better calibration.

**Claims And Evidence:**

Yes

**Requested Changes:**

Please the authors include more comparison with SOTA calibration methods and discuss the difference between SECE and [Soft calibration objectives for neural networks]

**Strengths And Weaknesses:**

The key problem of this paper is lack of comparison and discussion with related works:

For Gamma-Net providng sample-wise gamma values for focal loss, there are some focal loss based calibration methods such as [1][2], the author should compare with them. In addition, for the meta learning Gamma-Net itself, more motivation or empirical examples are needed.
For Smooth Expected Calibration Error (SECE), it is very similar with the [3], although it is compared in the experiments, it should be discussed in the main paper and related works, in terms of methodology.

---

[1] Dual focal loss for calibration

[2] Adafocal: Calibration-aware adaptive focal loss

[3] Soft calibration objectives for neural networks

---

> ### Author Response · Authors · 2024-04-24
> **Author Response to Reviewer mW6C**
>
> Thank you for your thoughtful suggestions. We have incorporated the recommended references and corresponding comparisons into Table 3. Additionally, we have included a discussion on the difference between [1] and our proposed Smooth ECE in both the Related Works section and the main content (Section 4.2, Discussion). The primary distinction lies in the utilization of a softmax-based bin-membership function by [1] to achieve differentiable calibration error. Differently, Smooth ECE, being an instance of non-parametric density estimators, circumvents the binning mechanism (opting instead for kernel-based approaches) and potentially mitigates binning bias [2], which we empirically evaluated in Section 5.2.2.
>
> [1] Karandikar et al., Soft calibration objectives for neural networks, NeurIPS 2021.
> [2] Zhang et al., Mix-n-Match: Ensemble and Compositional Methods for Uncertainty Calibration in Deep Learning. ICML 2020
>
> Thanks again for your helpful feedback and suggestions.

---

### Review · Reviewer_XKgH · 2024-04-17

**Summary Of Contributions:**

Training a classifier with focal loss has been known to improve the calibration of the network prediction. However, the gamma parameter in the focal loss plays a crucial role in making this approach effective. Motivated by a previous work [1] that used a meta-learning approach during calibration during training, this paper also proposes a meta-learning approach to learn a sample-wise gamma via a meta-network.


Reference:

[1] Bohdal et al., Meta-learning of model calibration using differentiable expected calibration error, 2021

**Audience:**

Yes

**Claims And Evidence:**

Yes

**Requested Changes:**

See weaknesses above

**Strengths And Weaknesses:**

Strengths

The work introduces a meta-network capable of learning sample-wise gamma in the focal loss, which results in competitive calibration performance to previous methods.

---

Weaknesses

1. The authors said the FLSD learns a sample-dependent schedule for gamma in focal loss. In the first line of the second page, the authors also mention that ‘different from FLSD, we learn a gamma parameter for each sample’. Could the authors clarify the difference?
2. To my understanding, $L_\text{SECE}$ is dependent on the output of the backbone model. Does this mean the $D_\text{val}$ is just the output of the backbone model?
3. Looking at Table 1, although the calibration performance is better in most cases, the proposed method tends to have the highest prediction error rate (i.e., lowest accuracy). The calibration error and accuracy are a trade-off in most cases, and it is not clear if the proposed method just prefers a better-calibrated prediction by sacrificing accuracy. Could the author provide a Preto front analysis to show that the proposed method achieves a superior solution rather than simply obtaining a prediction with lower calibration error and higher error rate?

---

> ### Author Response · Authors · 2024-04-24
> **Author Response to Reviewer XKgH**
>
> We would like to thank reviewer for the valuable feedback and suggestion. Following the reviewer's instructions, we have revised the submission to address the concerns outlined:
>
> > The explanation of the difference with FLSD [1].
>
> The main differences are: (1) FLSD is a non-learnable approach, it tries to find the solution for gamma value based on Lambert-W function (the Proposition 2 in FLSD [1]). With a chosen threshold (e.g., setting threshold to 0.2, gives $\gamma= 5$ for probability $\in [0; 0:2)$, and $\gamma= 3$ for probability $ \in [0:2; 1]$. The two $\gamma$ values are then scheduled across epochs. (2) Our approach is a learnable approach using an additional network, i.e., gamma-net. It learns a $\gamma$ value for each individual sample. We have revised paper to further stress this difference. Please refer to Related work and Section 4.1.
>
> > Question regarding the output of the backbone model.
>
> It is true that $L_{SECE}$ depends on the output of the backbone model, which takes the validation set ${D_{val}}$ as input and produces the extracted representations of samples in ${D_{val}}$ as output. These representations are subsequently utilized as input for the gamma-network, which is optimized by minimizing $L_{SECE}$.
>
> > The predictive-calibration trade-off and Preto front analysis.
>
> We appreciate the reviewer's suggestion to provide additional guidance. We fully agree that the calibration error and accuracy are a trade-off in most cases. Following your guidance, we conducted an analysis on the trade-off between predictive (measured by test error) and calibration errors (measured by ECE) based on Table 1. We added those content in the Appendix A.4. In general, we can see  that there exists a Preto front across many modern methods and there is no clear dominance between our method and all others. However, as depicted in Table 5 in the Appendix A.4, the ECE improvement is noticeable while the and accuracy loss for our proposed method, the reduction in predictive capability is modest; our method is superior to others when we consider a scalarized objective with equal weights. We also added this discussion in the revision.
>
> [1] Mukhoti et al., Calibrating Deep Neural Networks using Focal Loss. NeurIPS 2020.
>
> We appreciate reviewer's insightful review which, as you can tell, has allowed us to improve the paper. Please let us know if there is anything else we can do.

---

> > ### Comment · Reviewer_XKgH · 2024-04-28
> > **Response to Authors**
> >
> > I appreciate the authors for the detailed response. Especially, the addition of Appendix A.4 shows that the proposed solution tend to lie in the optimial pareto front. These have resolved my concerns.

---

### Decision · Action_Editor_2X3L · 2024-05-24

**Recommendation:** Accept with minor revision

**Comment:**

This work proposes a meta-learning approach for calibrating neural network uncertainty prediction, providing numerical evidence it approaches SOTA performance across different calibration metrics and tasks.

The initial reviews pointed towards insufficient numerical benchmarking and comparison to the literature, and a possible accuracy vs. calibration trade-off. During the rebuttal phase, the authors have clarified some points raised and update the manuscript, including further comparison to other methods in Table 3 and an additional section (Appendix A.4) following a reviewer's request.

The reviewers were overall positive with the revised version, except for an additional comment:

> *In the rebuttal, also, the authors state that "Error rate and classwise ECE measure (...) are not directly comparable." That remains an open question to me. For example, if the error rate is 10 in 100 samples, but the ECE, which is the L1 distance between prediction and accuracy is 1/100, then the calibration error is an order of magnitude lower than the error rate. In that case, what does an improvement in ECE mean? This trade-off is also discussed in Figure 7 of [1]. The rebuttal alludes to this point by saying, "The difference between SECE and DECE are indeed within one standard deviation apart in most cases, but we attribute it to the overall low values of calibration errors."*

Therefore, my recommendation is for the acceptance of the manuscript, conditioned on a minor revision to reflect the discussion above.

**Audience:**

Calibration of neural networks is a relevant topic of research with major implications to practice. Therefore, I expect this paper to interest a wide range of TMLR readers interested in uncertainty quantification.

**Claims And Evidence:**

This work proposes a meta-regularization approach to calibrate neural networks. The central claim is that the proposed algorithm achieves SOTA in different calibration metrics. This empirical claim is based on comparison accross different benchmarks, which has been judged insufficient by some of the reviewers. However, the authors were able to provide further evidence in their rebuttal, which was judged convincing by the reviewers concerned.